# Application and Experimental Validation of Seven-Degree-of-Freedom Beam Element for Girder Bridges during Deck Construction

Li Hui [1,*] , Md Ashiquzzaman [2] and Riyadh Hindi [3]

1 Civil Engineering Department, University of Louisiana at Lafayette, Lafayette, LA 70503, USA
2 Ameren Corp., St. Louis, MO 63103, USA; mashiquzzaman@ameren.com
3 Department of Civil, Computer and Electrical Engineering, St. Louis University, St. Louis, MO 63103, USA; riyadh.hindi@slu.edu
* Correspondence: li.hui@louisiana.edu

**Abstract:** During bridge deck construction, the deck finishing machine and the fresh concrete often produce large vertical loads and torsional moments acting on the bridge girder system. In some cases, these loads can cause excessive vertical deflection and transverse rotation in the bridge girders, leading to many maintenance and safety problems, such as changes in deck thickness and local and global instabilities during construction. To minimize the potential problems caused by deck construction, the AASHOTO LRFD Bridge Design Specification requires consideration of these torsional moments during the design procedure, and a detailed three-dimensional finite element analysis may be conducted. However, for bridge girders with open-section thin-walled sections, only the solid or shell element can be used to recognize the warping of the girder since the torsional warping effect is not included in the classical beam element. In this research, a warping degree of freedom was added to a beam element, and a three-dimensional beam element with seven degrees of freedom (7-DOF) at each node was derived as an alternative method for analyzing girder bridges during deck construction. A computer program based on the 7-DOF beam element was also developed in MATLAB. To assess the 7-DOF beam element, one bridge was selected to measure the transverse rotation, vertical deflection, and stress of the exterior girder and the first interior girder during deck construction. Also, three full-scale numerical models using solid elements, classical three-dimensional beam elements, and 7-DOF beam elements were created based on the geometries and loads of the experimental bridge. A comparative study was conducted by comparing the results from the numerical models and experimental monitoring data to evaluate the 7-DOF beam element. The results showed that the 7-DOF beam element had excellent behavior in analyzing the girder bridges under construction load, especially in the torsional analysis of bridge girders. Also, unlike the solid element model, which also provided reasonable results, the 7-DOF beam element model can compute the internal forces of the cross-sections along the bridge, which allows the 7-DOF beam element to be an alternative approach for design and research requiring less modeling effort and computational complexity.

**Keywords:** deck construction; torsional analysis; exterior girder rotation; warping effect; 7-DOF beam element

## 1. Introduction

Girder bridges are the most common type of small- and medium-span bridges, consisting of several similarly sized longitudinal girders spaced uniformly across the bridge's width. The deck slab is often designed to span transversely between the girders and extend past the exterior girder. This design increases the deck's width without the need for additional girders. Deck slab construction typically employs an overhang formwork system and a deck finishing machine. The overhang formwork is usually supported by steel brackets

placed at intervals of 90 cm (3 ft) to 120 cm (4 ft) along the exterior girder. These brackets support the weight of fresh concrete, the deck finishing machine, and other construction loads [1–3]. Figure 1 shows a typical formwork system and construction loads acting on bridge girders during deck construction. These overhang brackets are often attached to the top flange of the exterior girder and provide support against the bottom of the girder's web. The deck finishing machine, which creates significant loads during construction, is typically located on the edge of the overhang.

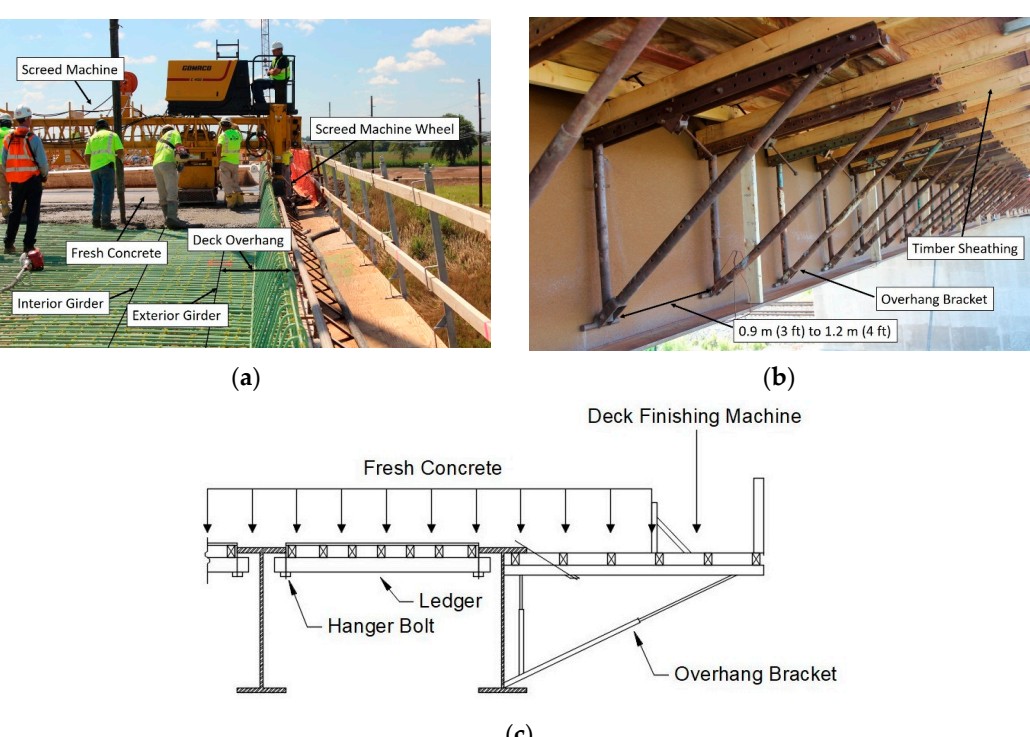

**Figure 1.** Typical formwork system and construction loads during deck construction. (**a**) Deck overhang and construction equipment. (**b**) Overhang bracket. (**c**) Loads at location of screed machine.

During bridge deck construction, a primary concern is the torsional moment exerted on the exterior girder [4–7]. The bridge shows less stiffness as the upper concrete deck slab does not contribute as effectively to the structure's rigidity. As the construction loads are transferred through the overhang bracket formwork system to the exterior girder, significant torsional moments are usually produced and sometimes can result in excessive transverse rotation of the girder [8]. Such excessive rotation can lead to an uneven deck thickness, causing maintenance challenges during the bridge's service life [8]. Additionally, local and global instabilities are potential issues, particularly for steel plate girders with slender webs when subjected to eccentric loads during deck construction [9–12]. These challenges can sometimes lead to bridge failures in steel and concrete bridge systems [8].

To mitigate potential issues arising from deck construction, the AASHTO LRFD Bridge Design Specification mandates the consideration of both lateral bending stress and the effects of bracket reactions during the design phase [13]. However, analyzing the lateral–torsional behaviors of bridge girders, especially those with open-section thin-walled beams like I-beams, remains challenging due to warping effects. The Guidelines for Steel Girder Bridge Analysis (2019) present a method that employs an equivalent torsion constant [14]. This method combines the St. Venant torsional stiffness for the open-section thin-walled beam, considering warping fixity at both ends of a specified unbraced length. A limitation of this approach is its assumption of the girder being simply supported, necessitating the calculation of the equivalent torsional constant for every variation in unbraced length or changes in the girder's cross-sectional properties.

Another common approach to account for warping effects utilizes the finite element method. The AASHTO LRFD Bridge Design Specification and The Guidelines for Steel Girder Bridge Analysis advocate for the finite element method. Specifically, a three-dimensional model with shell or solid elements is suggested to address warping torsion. This is because beam elements cannot recognize the torsional warping degree of freedom [14]. However, implementing finite element modeling with shell or solid elements requires extensive expertise in structural analysis and significant modeling efforts.

To incorporate the warping effect in the analysis of structures with thin-walled open sections, V.Z. Vlasov formulated a torsional theory that includes the impact of restraint warping [15]. In this torsional theory, the warping constant and bi-moment were introduced to account for torsional restraint warping, sectional deformation, and the influence of shear strain on normal stress distribution. Moreover, several studies have been conducted to provide a deeper understanding of the mechanical behavior associated with torsional warping [16–23]. Specialized finite beam elements have been developed for specific applications based on the torsional warping theory. S.H. Zhang et al. introduced a beam element that accounted for the impacts of warping, distortion, and shear lag in box beams [24]. This element features two nodes, each with nine degrees of freedom. Later, in 2017, F. Cambronero-Barrientos proposed a three-dimensional, three-node beam element [25]. It has five degrees of freedom per node and is designed to incorporate effects like shear lag, torsion, distortion, and both homogeneous and non-homogeneous distribution of normal stress.

In addition, various general-purpose beam elements accounting for warping effects have been developed and incorporated into several finite element software packages (e.g., ABAQUS/CAE [26], ANSYS [27], and ADINA [28]). Typically, these software solutions introduce a seventh degree of freedom (7-DOF) at nodal points to encompass the warping effect. However, in certain scenarios, the analysis executed by these software tools might provide inconsistent results, particularly for beams undergoing non-uniform torsion [29]. Expanding on this, Sabat and Kundu explored advanced modeling techniques in ANSYS for complex torsional analysis, enhancing the accuracy of predictions [30]. Similarly, Li et al. proposed improvements in warping analysis in ABAQUS, focusing on the shear lag effect for a thin-walled box-section beam [31].

Some other researchers also developed beam elements for general purposes. For instance, E.J.Sapountzakis formulated a beam element with a $14 \times 14$ stiffness matrix based on the boundary element method (BEM) [32]. This beam element accounted for warping and shear deformation effects, catering to nodal load vectors of any given homogeneous or composite cross-section. Moreover, Yau and Kuo introduced a new $14 \times 14$ stiffness matrix for a beam element to analyze the warping effect on the buckling of the I-beams [33]. Their approach leaned on the Vlasov torsional theory and operated under the assumption that the I-beam was perfectly straight, constructed from three slender flat plates, and that the section remained bi-symmetric. Bernuzzi and Simoncelli investigated the interaction between axial forces, bending moments, and bi-moments in the linear elastic range, especially focusing on members with mono-symmetric cross-sections. Their research evaluated the impact of various transformation matrices on analysis outcomes, highlighting how the selection of these matrices significantly affects the results [34]. Furthermore, extensive research has been undertaken to innovate numerical analysis methods for beam warping analysis, as evidenced by several studies in this field [35–39].

In this study, a three-dimensional beam element with seven degrees of freedom (7-DOF) was derived, grounded in the kinematics of beams under non-uniform torsional loads and the theory of potential energy. Compared with the classical three-dimensional beam element with six degrees of freedom in each node, the twist angle per unit length was taken as an additional degree of freedom to represent the torsional warping effect. Based on the proposed 7-DOF beam element, a finite element computer program was developed in this study. In order to evaluate the 7-DOF beam element in analyzing the bridge girders during deck construction, one bridge located in the state of Illinois (USA) was monitored

and finite element analyses were conducted using different elements. Four finite element models were established using a solid element, a classical beam element, a B32OS element from ABAQUS/CAE, and the 7-DOF element. The finite element results were compared with the experimental data, and they showed that the 7-DOF frame element developed in this study yielded accurate and reliable results. Additionally, the 7-DOF beam element achieved this with significantly reduced modeling effort and computational time compared to other methods.

## 2. The 7-DOF Beam Element

The major difference between the 7-DOF beam theory and classical beam theory is the assumption of the warping effect for the beam section. The classical beam theory basically assumes that a cross-section orthogonal to the *x*-axis always remains plane and keeps its shape during deformation. It adopts the St-Venant torsional theory, which assumes that the cross-section is free of warping or the warping along the beam is constant [40,41]. However, for an open thin-walled cross-section such as an I-beam that is widely used in structural design and construction, the warping effect is often dominant during the torsional analysis, especially when the beam is under significant torsional load [25].

The 7-DOF beam element considers warping deformation as an additional degree of freedom during the analysis. Generally, the rotation caused by the torsional moment reduces because the torsional moment is resisted by both St-Venant torsional stress and warping stress. A typical 7-DOF beam element is shown in Figure 2.

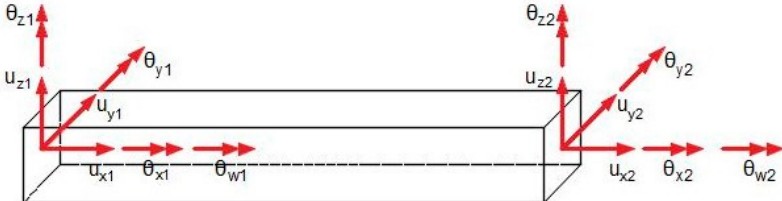

**Figure 2.** The 7-DOF beam element.

The force applied at each element and the corresponding deformation are written as Equations (1) and (2):

$$F = \left[ F_{x1} F_{y1} F_{z1} M_{x1} M_{y1} M_{z1} M_{w1} F_{x2} F_{y2} F_{z2} M_{x2} M_{y2} M_{z2} M_{w2} \right]^T \tag{1}$$

$$U = \left[ u_{x1} u_{y1} u_{z1} \theta_{x1} \theta_{y1} \theta_{z1} \theta_{w1} F_{x2} F_{y2} F_{z2} \theta_{x2} \theta_{y2} \theta_{z2} \theta_{w2} \right]^T \tag{2}$$

where $M_w$ is the warping moment and $\theta_w$ is the angle of twist per unit length.

### 2.1. Deformation and Kinematics of 7-DOF Beam Element

When the beam is restrained at one or more locations along the span, the section no longer remains flat, and additional deformation in the axial direction is induced. Figure 3 shows the general beam deformation in each coordinate plate.

By adding warping-induced axial deformation, the deformation on each coordinate axis can be written as Equations (3)–(5):

$$u_x(x, y, z) = w_x(x) + z\theta_x(x) - y\theta_z(x) + \omega(y, z)\theta_{x,x} \tag{3}$$

$$u_y(x, y, z) = w_y(x) - z\theta_x(x) \tag{4}$$

$$u_z(x, y, z) = w_z(x) + z\theta_x(x) \tag{5}$$

where $u_x$, $u_y$, and $u_z$ are the accumulated deformation in the *x*, *y*, and *z* directions, respectively, and $\omega(y, z)$ is the warping function, which basically is a shape function defining the axial deformation of the cross-section from the rotation component.

As a result of including warping, the strain at a beam cross-section can be described with the following equations (Equations (6)–(8)):

$$\varepsilon_{xx} = u_{x,x} = w_{x,x} + z\theta_{y,x} - y\theta_{z,x} + \omega(x,y)\theta_{x,xx} \tag{6}$$

$$\gamma_{xy} = u_{x,y} + u_{y,x} = w_{y,x} - \theta_z + (\omega_{,y} - z)\theta_{x,x} \tag{7}$$

$$\gamma_{xz} = u_{x,z} + u_{z,x} = w_{z,x} + \theta_y + (\omega_{,z} + y)\theta_{x,x} \tag{8}$$

Equations (6)–(8) indicate that the warping of the beam section not only produces additional axial strain but also induces extra shear strains on the cross-section. The stress at a certain point of the cross-section can be simply computed based on Hook's Law.

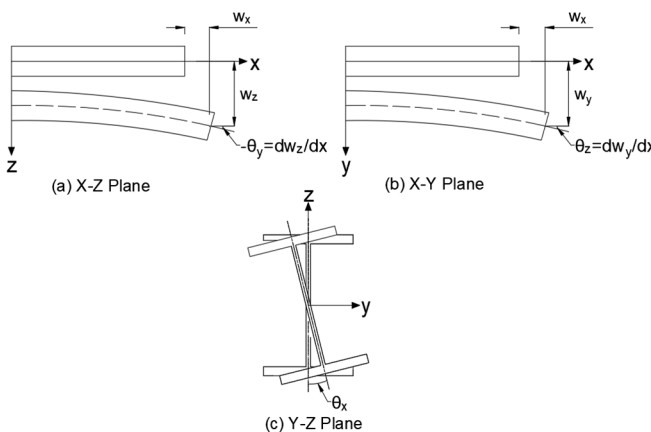

**Figure 3.** Deformation field in beam element.

### 2.2. Potential Energy and Stiffness Matrix

To derive the stiffness matrix for the 7-DOF beam element, the potential energy principle was applied. When external forces are applied on the beam element and perform work, the energy is stored in the form of stress and elastic deformation [42,43]. This means the work done by the external forces must be equal to the potential energy stored in the beam. The total potential energy of a linear elastic body can be defined using Equation (9):

$$\Pi = U - W \tag{9}$$

where $U$ is the strain energy and $W$ is the total work done by the applied forces, which can be written as Equations (10) and (11):

$$U = \frac{1}{2}E\int_V \varepsilon_{xx}{}^2 dV + \frac{1}{2}G\int_V \gamma_{xy}{}^2 dV + \frac{1}{2}G\int_V \gamma_{xz}{}^2 dV \tag{10}$$

$$W = F^T \times U \tag{11}$$

For the conservative structural system, the kinematically admissible deformations that correspond to the equilibrium state extremize the total potential energy. If the extremum is a minimum, the equilibrium state is stable [44]. Therefore, the minimum potential energy (MPE) can be used to find relations between the applied forces and deformations, which generates a stiffness matrix. The MPE for the 7-DOF beam system can be described using Equation (12).

$$\frac{\partial \Pi}{\partial U} = 0 \tag{12}$$

By solving the MPE, the relation between the applied force and displacement can be written as Equation (13).

$$[F] = [K^e][u] \tag{13}$$

The element stiffness matrix $K^e$ is a 14 by 14 symmetric square matrix.

## 3. Field Bridge Experiment

One bridge located in Lawrence County (Lawrence County, IL, USA) was selected for this research to validate the 7-DOF frame element model. The instrumented bridge is a three-span continuous bridge with six plate girders. The overall span length of the bridge is 157 m (516 ft). The steel plate girders are 167.64 cm (66 in) in depth with 50.8 cm (20 in) wide flanges and are spaced at 2.18 m (86 in). The overhang width is 1.12 m (44 in). A framing plan of the bridge, girder elevations, and details of the cross-frames is shown in Figures 4–6.

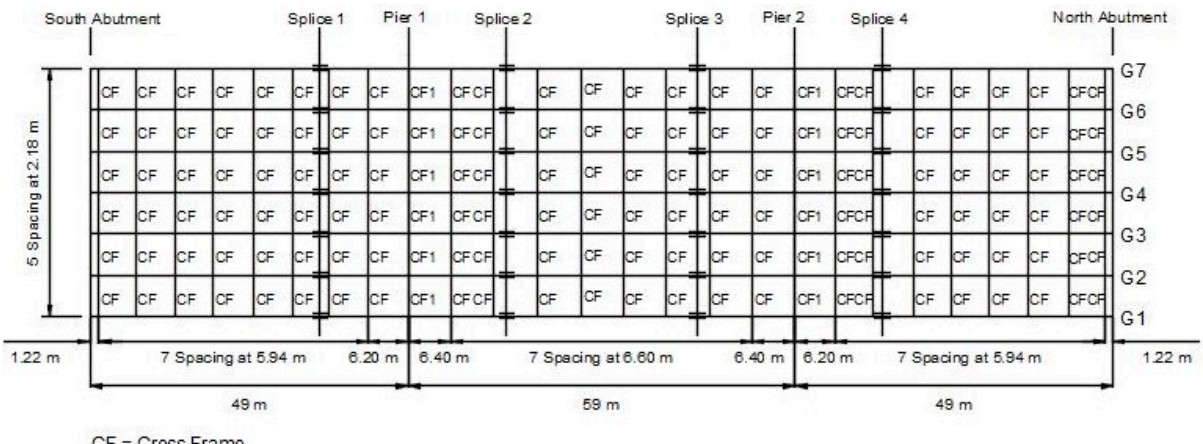

**Figure 4.** Framing plan of the bridge.

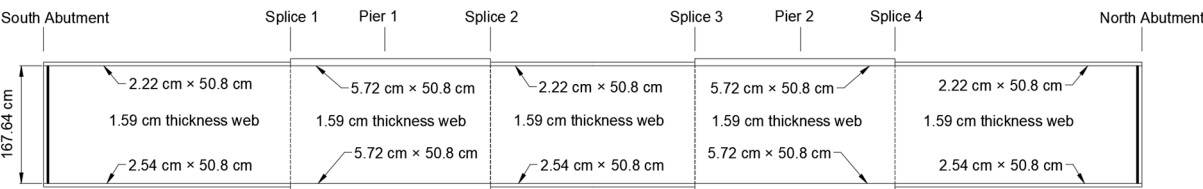

**Figure 5.** Girder elevation of the bridge.

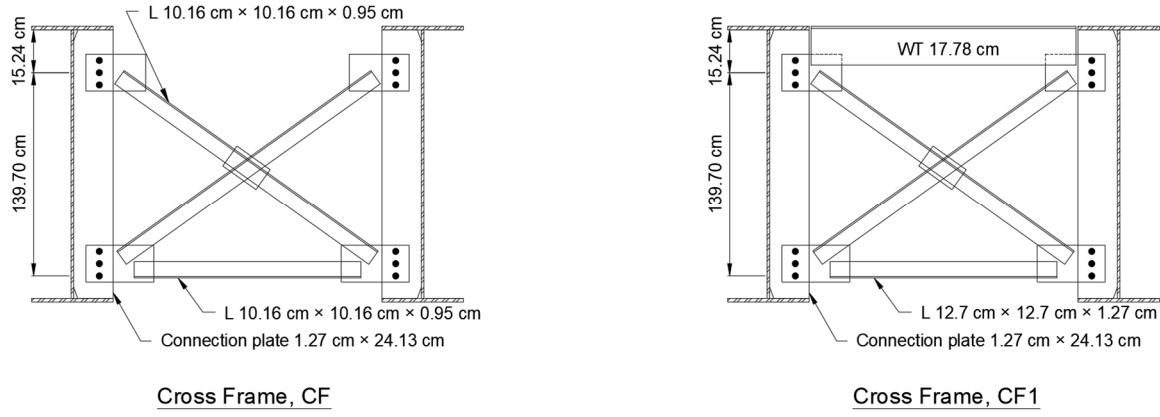

**Figure 6.** Cross-frame of the bridge.

The deck pouring of the experimental bridge was divided into two sequences. As shown in Figure 7, the first sequence covered from both abutments to 34.4 m (112.75 ft) along the side spans and the middle portion of the second span. In this stage, the screed machine moved from the south abutment to the north abutment along the bridge span. The second deck pour placed the fresh concrete on the rest of the bridge three days after the

first pour, with the screed machine moving in the same direction. The fresh concrete was poured at night for both sequences to control early cracking due to temperature effects.

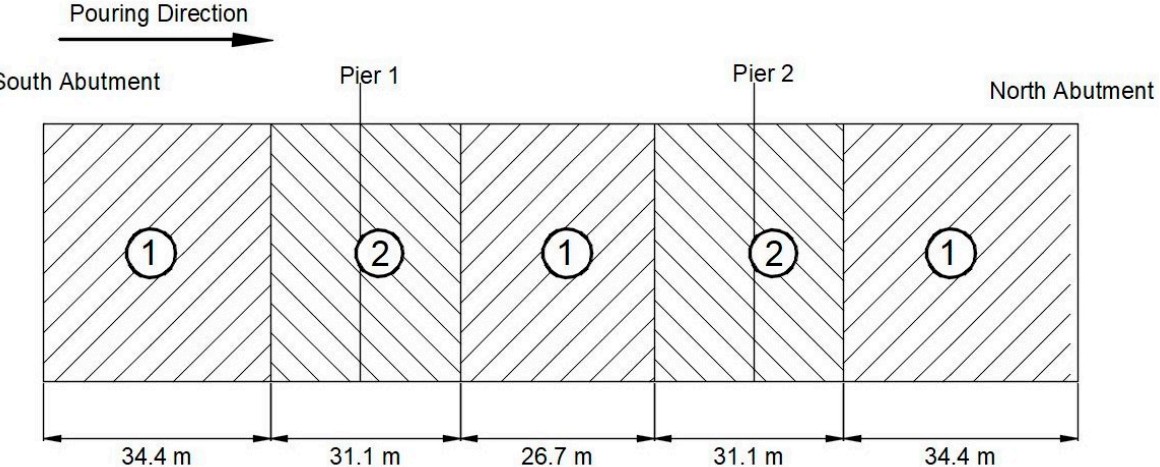

**Figure 7.** Deck pouring sequence of the bridge.

*3.1. Bridge Instrumentation*

The exterior girder (G1) and first interior girder (G2) of the first span of this bridge were instrumented and monitored for deformation and strain during the first concrete deck pour in the southern side span (shown in Figure 8). Three sections (S1, S2, and S3) were identified for the installation of the sensors. At each section, tilt sensors were installed on the middle and bottom of girder web on both exterior and interior girder. Prisms were installed at the bottom flange of each section in order to measure the deflection of the girders using total stations. Two stain gauges were also placed on the bottom flanges in S2, as shown in Figure 9.

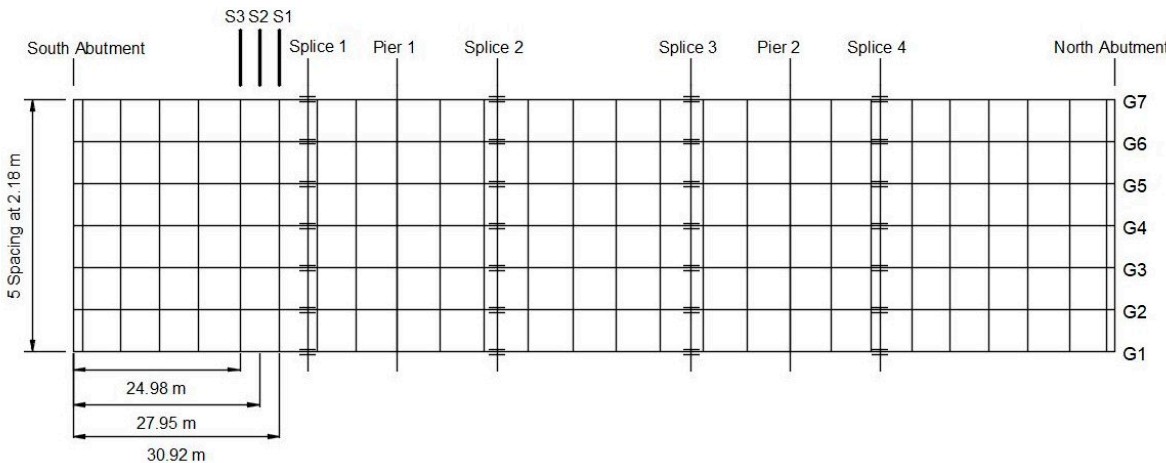

**Figure 8.** Predefined sections for instrumentation.

The transverse rotation of the bridge girders was measured using dual-axial (CXTLA02) tilt sensors. The tilt sensors have a twenty-degree maximum range in both directions. The sensitivity is 1% and the resolution is 0.03 deg rms. Also, an open aluminum box was used to hold the tilt sensor for protection, as shown in Figure 10a. Foil strain gauges (CEA-06-125UN-350) were installed on the bottom of the girder flange to measure the strain changes induced by deck construction loads. The strain gauges have a sensitivity of $350 \pm 0.3\%$ ohms, a resolution of 1 µm/m, and a range greater than 5000 µm/m. A protective coating was also used to avoid the effect of moisture and protect the strain gauges from damage (Figure 10b). The deflection of the bridge girders during deck construction was measured using the total station, as shown in Figure 11.

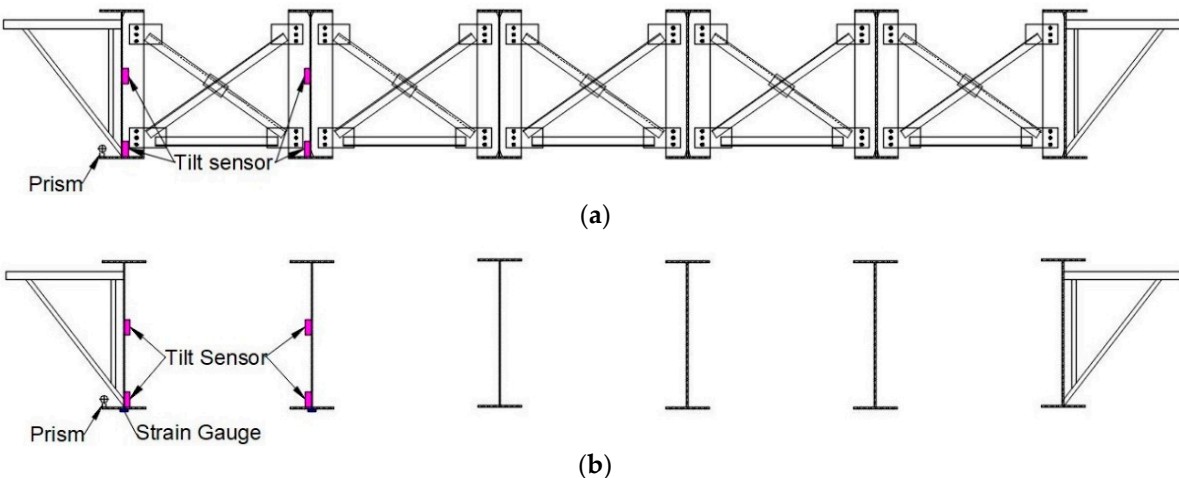

**Figure 9.** Location of tilt sensors, strain gauge, and prism for the experimental bridge: (**a**) location of sensors at S1 and S3, (**b**) location of sensors at S2.

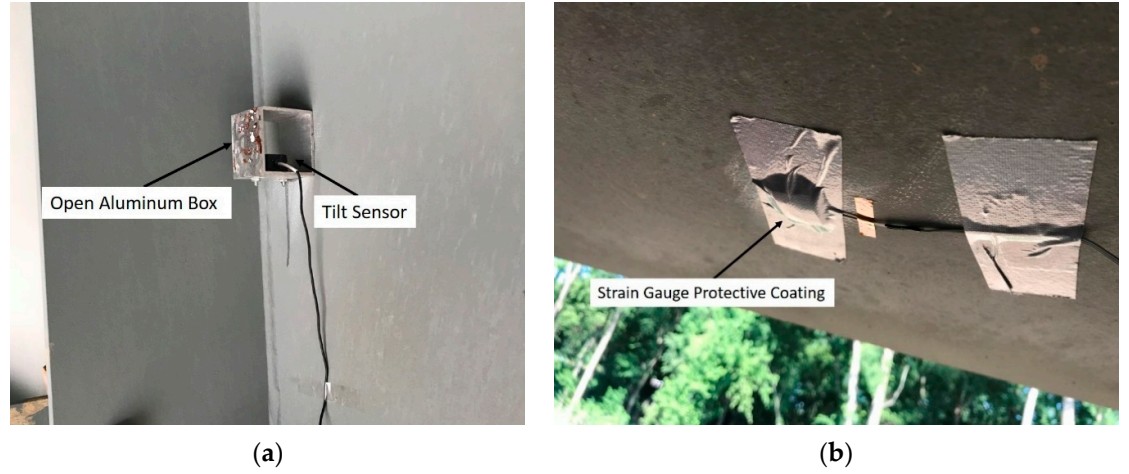

**Figure 10.** Tilt sensor and strain gauge: (**a**) Tilt sensor with open aluminum box. (**b**) Strain gauge with protective coating.

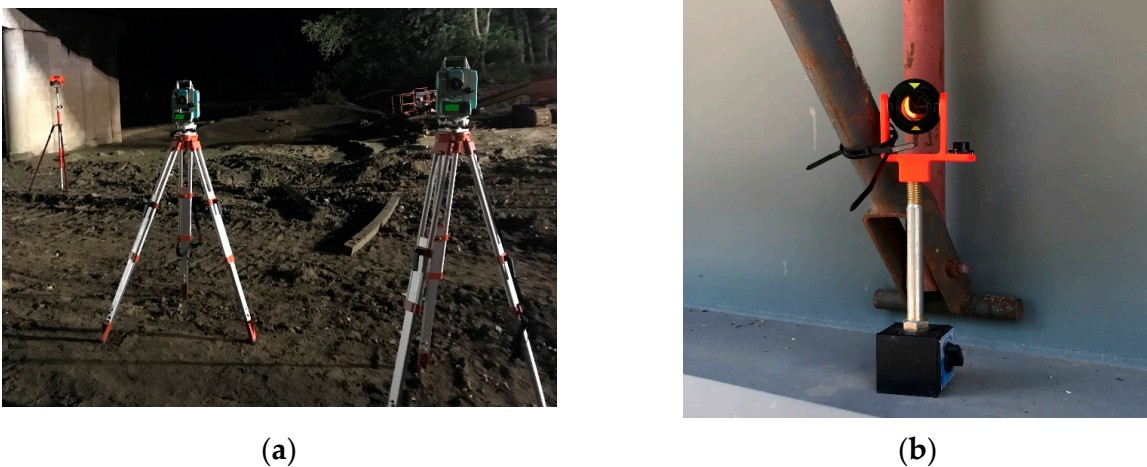

**Figure 11.** Total station and prism: (**a**) Total station. (**b**) Prism.

### 3.2. Experimental Results

The readings of sensors were collected at thirty-second intervals during the first pouring sequence of the experimental bridge. The tilt sensors measured both the inward (positive) and outward (negative) transverse rotation of the beam. However, due to the small measuring interval and the vibration of the screed machine on the top of the bridge, the experimental data from tilt sensors showed fluctuations during the whole monitoring period, making it difficult to recognize the changes in transverse rotation that occurred during deck construction. Therefore, the exponential moving average (EMA) method, a weighted moving average method used for smoothing time-series data, was used to reduce these fluctuations in the experimental data.

The EMA method was also used to calculate the value of rotation. Generally, the maximum rotation for a specific section occurred when the screed machine and fresh concrete moved right at the section. Figure 12 shows the exterior girder rotation at the bottom of the web in section S1. The results indicate that a maximum rotation value of 0.08 degrees occurred when the screed machine moved above section S1.

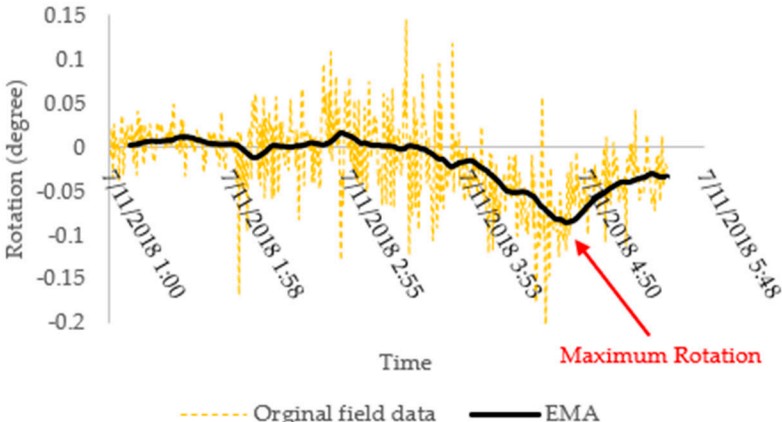

**Figure 12.** Exterior girder rotation at the bottom of the web on section S1.

The collected strain data from the bottom of the G1 and G2 girders was analyzed. Figure 13 illustrates the stress time-series data, indicating the behavior of the bridge girders, specifically the bottom of both the G1 and G2 girders, throughout the deck construction phase. It is observed that G1-Bottom experienced a series of fluctuations before establishing a steady increase, closely mirroring the trend exhibited by G2-Bottom. This behavior can be attributed to the fact that G1, being an exterior girder, endures more torsional moments during the deck construction. As the screed machine operated over the S1 section, the stress values peaked at 61.7 MPa (8.95 ksi) and 47.2 MPa (6.85 ksi) for the G2 girders.

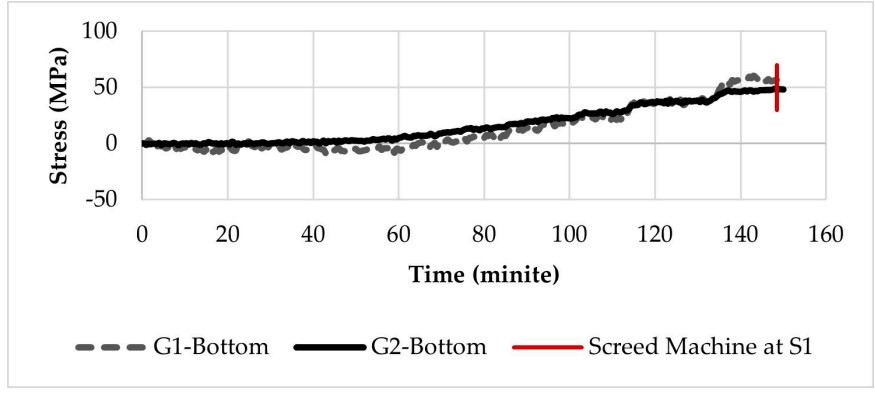

**Figure 13.** Strain gauge data at section S2.

Figure 14 illuminates the time-dependent vertical deflection profiles for three distinct sections: S1, S2, and S3. Similar to the stress and rotation, peak deflection is noted when the screed machine is directly positioned over the S1 section. The recorded deflections for the sections are 5.93 cm (2.33 inches), 7.02 cm (2.76 inches), and 7.68 cm (3.02 inches), respectively.

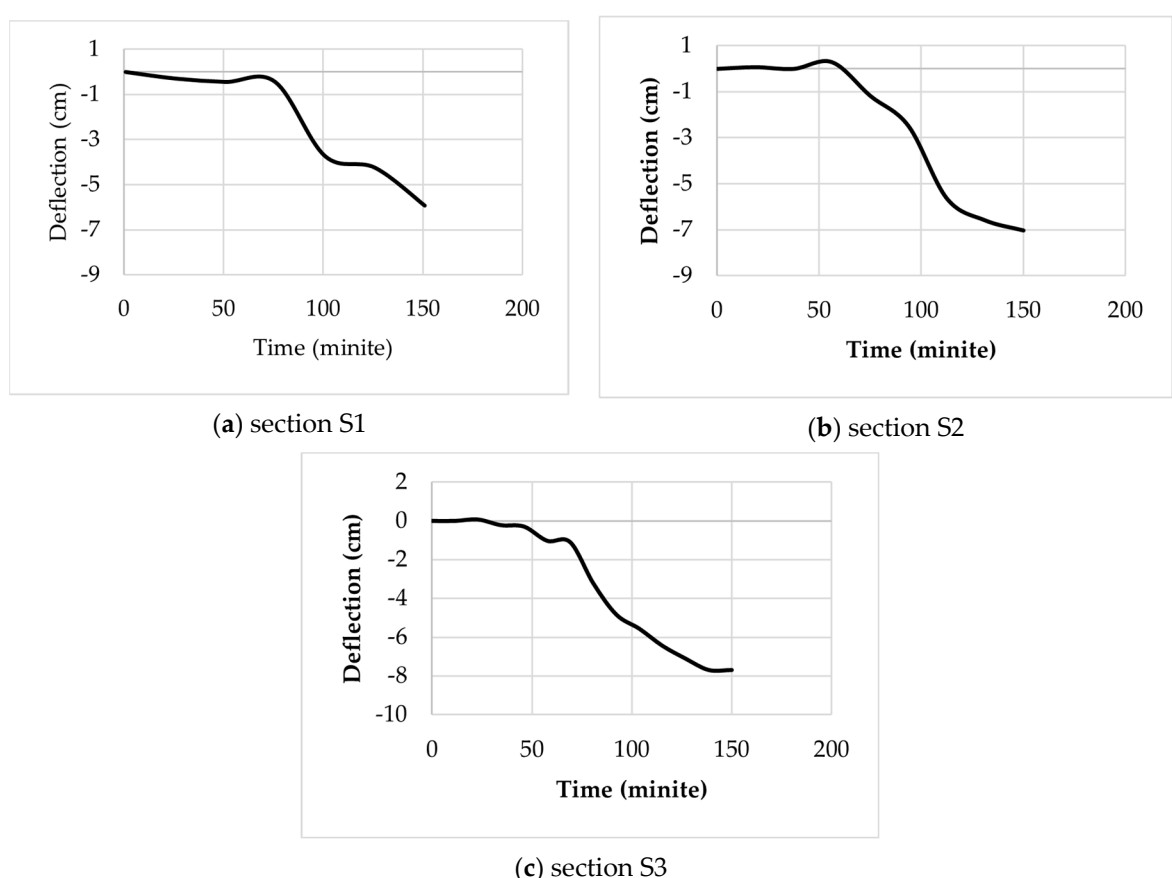

(**a**) section S1

(**b**) section S2

(**c**) section S3

**Figure 14.** Vertical deflection at (**a**) section S1, (**b**) section S2, and (**c**) section S3.

## 4. Numerical Modeling

Three finite element models using solid elements, classic beam elements, and 7-DOF beam elements were created and compared with the in-field measurement to evaluate the accuracy of these models. To ensure that all three models were comparable, the loads applied in the models were based on where the maximum transverse rotation occurred in section S1. In this case, the fresh concrete covered from the south abutment to the location of section S1, and the screed machine was right at section S1. Figure 15 shows the location of the fresh concrete and screed machine used in the finite element analysis. The mesh size of the girders was determined to be six inches for all three models.

The solid element model for the experimental bridge was developed in ABAQUS/CAE. The C3D8R element, which is widely used in stress and displacement analysis, was used in the modeling of bridge girders. The cross-frames were modeled using the C3D10 element due to the irregular shape and connection of the components. The C3D10 element is a second-order tetrahedral three-dimensional stress element with ten nodes in each element, which can generate meshes for complex geometry. Figure 16 shows the full-scale finite element model for the experimental bridge in ABAQUS/CAE. The bridge girders and cross-frame were modeled based on the design dimensions. A single-angle K-frame without a top truss was used in this bridge at all non-support locations (shown in Figure 17a). For the support location, an X-frame was used with both top and bottom trusses (shown in

Figure 17b). Welded connections between the girder webs and connection plates of the cross-frames were simulated using the tie connection.

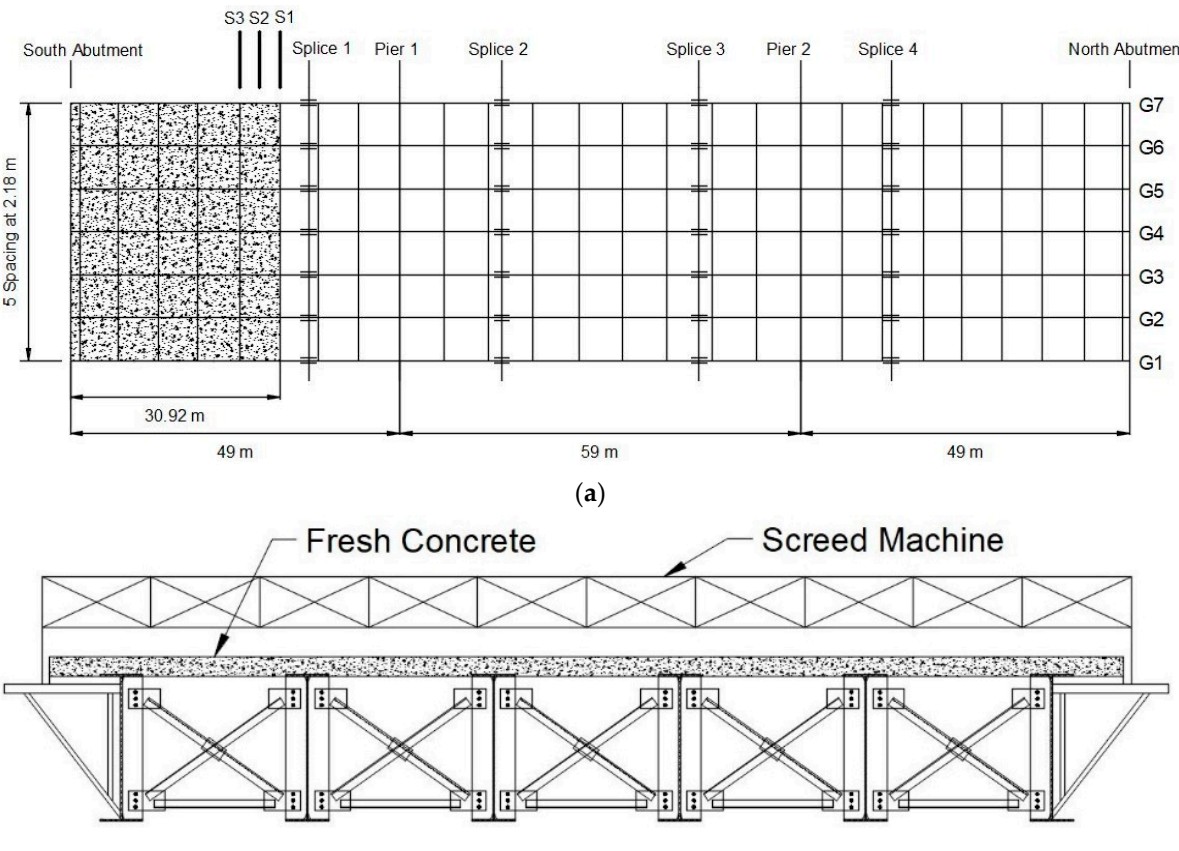

(**a**)

(**b**)

**Figure 15.** Location of fresh concrete and screed machine used in finite element analysis: (**a**) fresh concrete location; (**b**) loads at section S1.

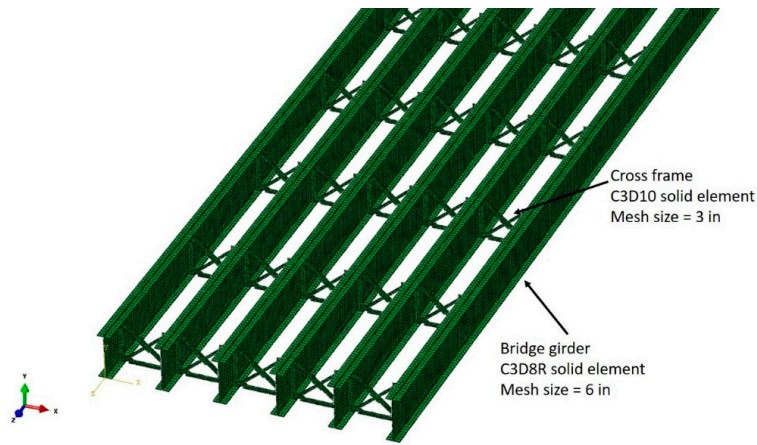

**Figure 16.** Solid element model created using ABAQUS/CAE.

The classical beam element model was created using the SAP2000 v14 software. All the structural components including girders and cross-frames were modeled using classical three-dimensional beam elements. In order to obtain a better comparison, the cross-frames were modeled based on the same geometrical dimensions as used in the solid element model. Rigid beam elements were also used to connect beams to cross-frames to maintain the same cross-frame depth as specified in the design. Figure 18 shows the classical beam element model that was built in SAP2000.

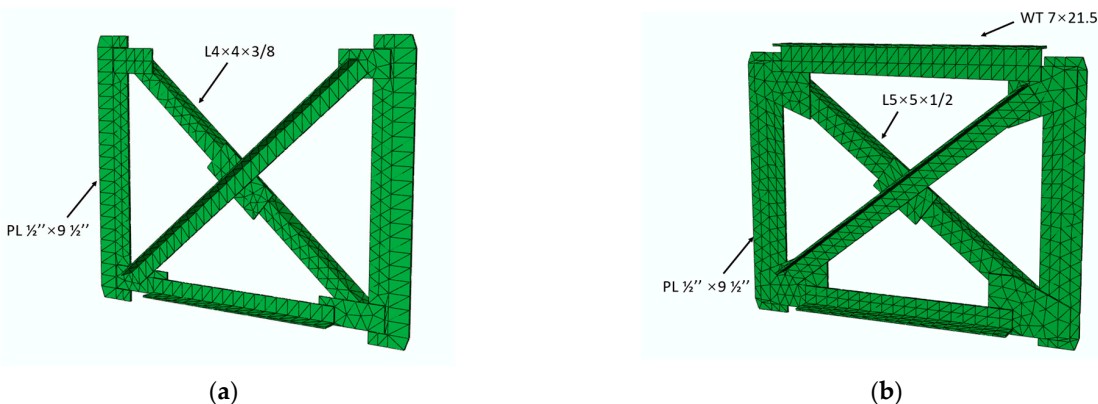

(**a**)                                        (**b**)

**Figure 17.** Cross-frame modeled in ABAQUS/CAE: (**a**) cross-frame at non-support location; (**b**) cross-frame at support locations.

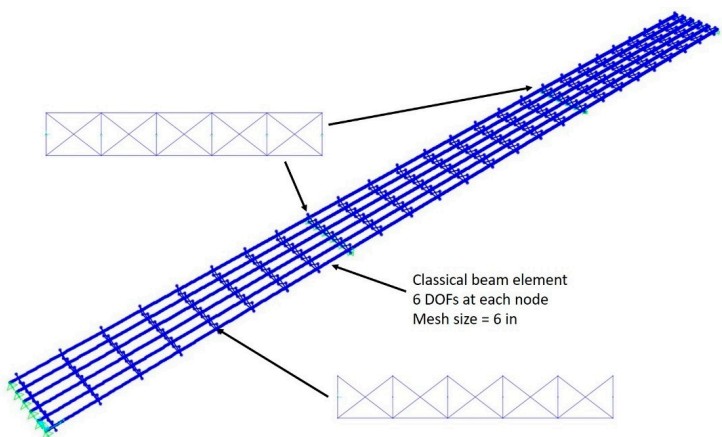

**Figure 18.** Classical beam element model built in SAP2000.

A finite element computer program was developed in MATLAB in order to evaluate the 7-DOF beam element developed in this study. The general procedures for finite element analysis using 7-DOF beam elements are shown in Figure 19. The program required the input of the basic bridge information such as girder dimensions, span lengths, overhang width, girder spacings, details of lateral bracing and mesh size, etc. Then, all the information for generating local and global stiffness matrices was created automatically. The local stiffness matrix and coordinate transformation matrix were 14 × 14 matrices. The size of the global stiffness matrix was larger than that in the classical beam element model but much smaller than the solid element model, depending on the total number of nodes. The program outputs included deformations, internal forces, reaction forces, and stresses.

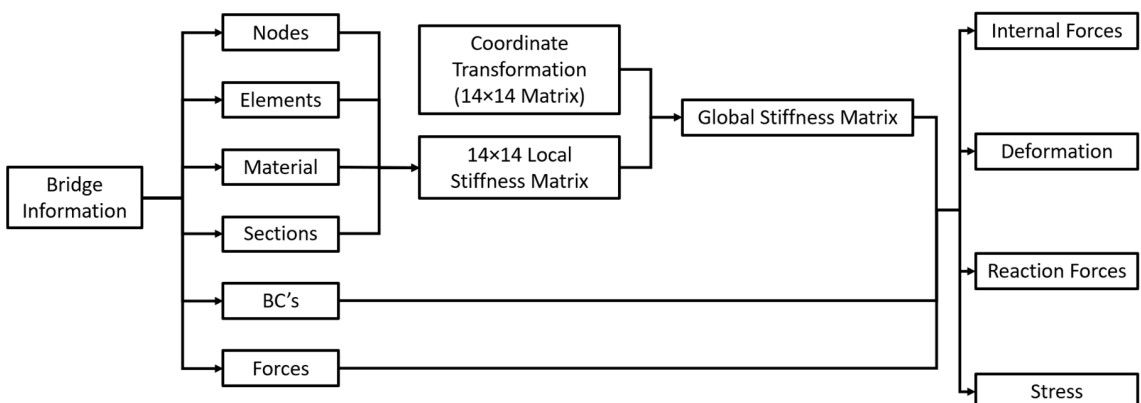

**Figure 19.** Programming procedure for 7-DOF frame element model.

The 7-DOF beam element model was similar to the classical beam element model. These two models were able to share the same node coordinates and element information. However, much attention must be paid when generating the force vector and boundary conditions since the warping degree of freedom was added to the stiffness matrix. The 7-DOF beam element created by MATLAB is shown in Figure 20. Both the girders and cross-frames were modeled using the 7-DOF beam element developed in this study with a mesh size of six inches.

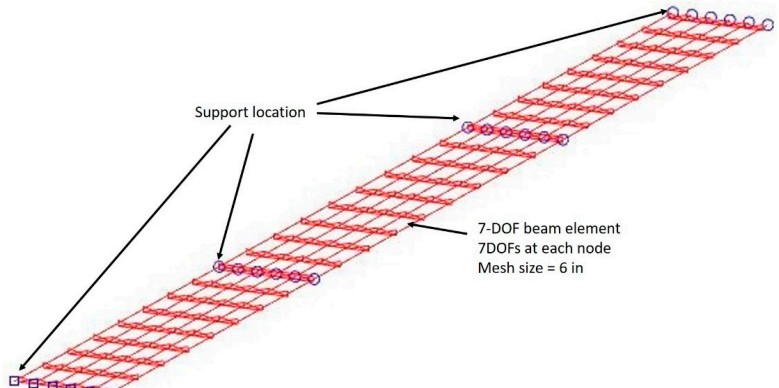

**Figure 20.** 7-DOF beam element model created in MATLAB.

The loads applied on the 7-DOF beam element model were the same as those on the classical beam element model since they were beam elements in both cases. In terms of programming, the analysis using the 7-DOF beam element did not require a total rewrite of the finite element program since only minor modifications were required for the stiffness matrix, boundary conditions, and force vectors.

## 5. Results

### 5.1. Rotation and Deflection

The torsional behaviors of the bridge girder are considered the major difference between the classical and 7-DOF beam element models. Figure 21 shows a comparison of transverse rotation on the exterior and first interior girder among three numerical models and experimental data in section S1. The experimental data and solid element model indicated that the middle and bottom of the girder web had very similar transverse rotations for both the exterior girder (G1) and the first interior girder (G2). The 7-DOF element model and solid element model showed very close transverse rotation for the exterior girder—0.087 and 0.073 degrees, respectively, which agreed with the experimental data of 0.087 degrees for the middle of the web and 0.080 degrees for the bottom of web. However, the classical beam element model provided a rotation value of 0.46 degrees for the exterior girder, which was around six times larger than the experimental data and 7-DOF beam element model. Only minor differences were observed among the finite element models and experimental data for the interior girder. This is because no torsional moment was applied to the interior girder, and the rotation resulted from relative deflections among girders as well as the bending of the bracing systems.

A comparison between rotations from the numerical models and experimental data for section S2 is shown in Figure 22. The classical beam model shows a significantly large rotation value of 3.16 degrees for the exterior girder, which is twenty-six times larger than the experimental data (0.12 degrees). This is due to the absence of bracing in section S2, and the classical beam element cannot account for the warping constraint at the adjacent bracings. Furthermore, the 7-DOF beam element and solid element model observed rotations of 0.10 and 0.11 degrees, respectively, which are reasonably close to the experimental data, with a difference of less than 15%. For the interior girder, both the numerical analysis results and experimental data showed rotations of less than 0.10 degrees, as no torsional moment was directly applied to the interior girder.

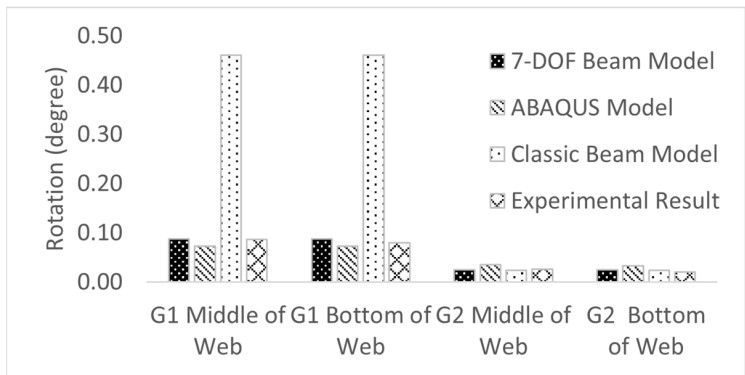

**Figure 21.** Rotation comparison among different numerical models and experimental data for S1 section.

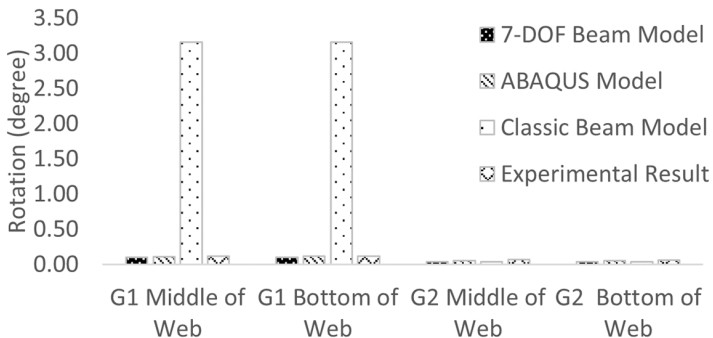

**Figure 22.** Rotation comparison among different numerical models and experimental data for S2 section.

Figure 23 shows the rotations of both the exterior and the first interior girder in section S3. The 7-DOF beam model yielded a rotation value of 0.09 degrees, which was comparable to the results from the solid element model and experimental data, at 0.11 degrees and 0.12 degrees, respectively. However, the rotation value from the classical beam element model was notably off, showing 0.50 degrees, which is 500% greater than both the 7-DOF beam element result and the experimental data. For the interior girder in section S3, the 7-DOF beam element model indicated a rotation of 0.06 degrees, showing less than a 15% difference when compared to the solid element model and experimental data, both at 0.07 degrees. However, a 40% difference was observed when comparing the results from the classical beam element model to the others.

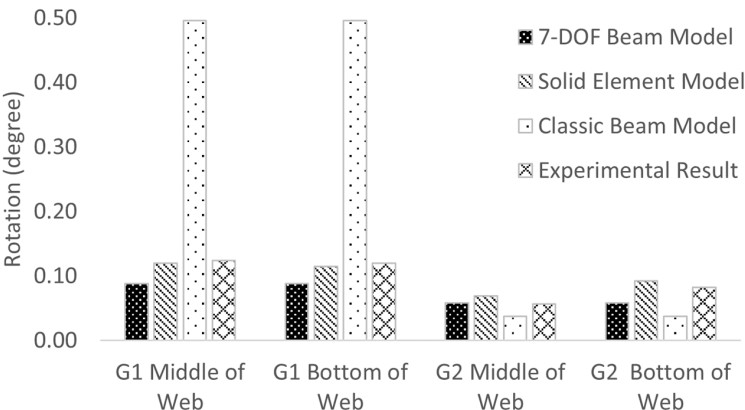

**Figure 23.** Rotation comparison among different numerical models and experimental data for S3 section.

The comparison among numerical models, which use the 7-DOF beam and other elements, as well as experimental data, demonstrates that the 7-DOF beam element is capable of analyzing the torsional behaviors of girder bridges with thin-walled open sections during deck construction. The transverse rotations observed for both the exterior

and the first interior girders across three different sections were reasonable when assessed using the 7-DOF beam element. Moreover, using the 7-DOF beam element requires less computational time and effort in modeling.

Figure 24 presents a comparison of vertical deflection among the numerical models and experimental data. Since warping stiffness does not significantly influence the bending behaviors of the bridge girder, the 7-DOF beam element model exhibits nearly identical deflections to the classical beam element model in the vertical direction across all three sections. Additionally, only slight differences are observed between the 7-DOF beam element and the solid element model, with the maximum variation being 2.3%. The experimental data are approximately 30% lower than the results derived from the 7-DOF beam element. This discrepancy can be attributed to the numerical model's oversight of the added stiffness provided by the formwork system. In general, the 7-DOF beam element aligns with the other numerical methods when analyzing vertical deflection.

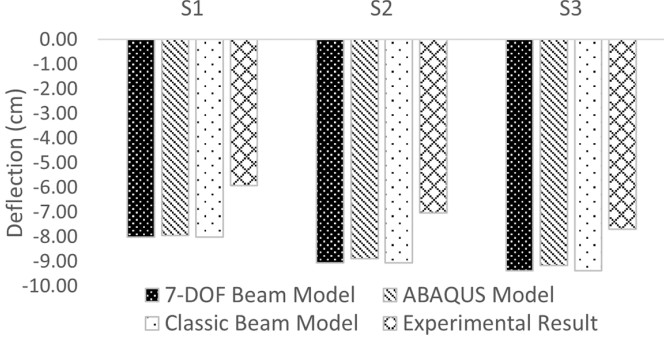

**Figure 24.** Vertical deflection comparison among different finite element models for S3 section.

### 5.2. Stress

As shown in Figure 25, the stress analyzed using the 7-DOF beam element was compared with the other numerical models and experimental data at section S1, where the strain gauges were installed on the bottom flange of the exterior girder (G1) and first interior girder (G2). Stresses of 63.9 MPa (9.27 ksi) in the exterior girder and 61.7 MPa (8.95 ksi) in the interior girder were computed using the 7-DOF beam element. The solid element and classical beam element models gave similar stress values with less than a 1% difference compared with the 7-DOF beam element model. The stresses collected from the experimental bridge were slightly smaller than the numerical analysis result. The differences between the 7-DOF beam element model and experimental data were 4% and 20% for the exterior girder and interior girder, respectively. It is also important to note that the warping effect can change the stress distribution on the cross-section according to Equations (6)–(8). For the experimental bridge, since the primary stress on the cross-section was due to the bending of the girder, the warping stress had a very limited effect on the stress calculation. However, if the bridge girder is under a large torsional moment, the stress induced by the warping effect can significantly affect the stress distribution of the cross-section.

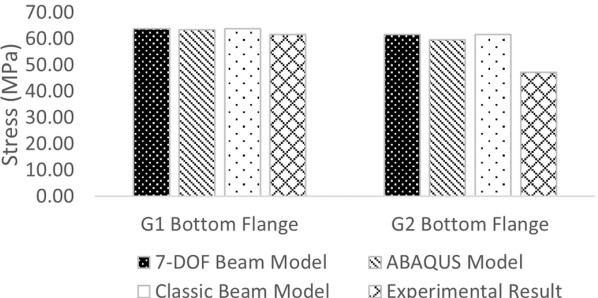

**Figure 25.** Stress comparison among different finite element models.

## 6. Discussion

This comparative study confirmed the accuracy of the 7-DOF beam element in analyzing girder bridges during deck construction. This element stands out for its inclusion of warping as an integral degree of freedom, enhancing the realism and accuracy of structural simulations. This approach marks a significant deviation from traditional methodologies employed in other studies, which often focus on warping beam elements.

The purpose of the 7-DOF beam element in this study is to provide a method for the detailed analysis of complex structural behaviors, especially in scenarios involving girder bridges. It includes a comprehensive set of degrees of freedom, encompassing warping to simulate real-world structural responses more accurately. This is in contrast to the warping beam elements used in other studies, which are primarily designed to address warping deformations in structural members and are less sophisticated.

In terms of accuracy, the 7-DOF beam element demonstrates a high degree of precision, particularly in predicting torsional behavior and warping effects. Its advanced formulation allows for a more detailed representation of real-world conditions, which is a notable improvement over the simpler warping beam elements of other studies. These traditional warping elements, while effective for modeling warping deformations, may not capture the full spectrum of structural behaviors, especially under complex loading conditions.

Validation was achieved by comparing the results from the 7-DOF beam element model with those from the solid element models and experimental data. The classical beam element was found to be inadequate in accurately predicting the torsional behavior of the bridge under torsional moments. This inadequacy largely stemmed from the classical beam element's assumption that the girder experienced no restrained warping. In contrast, the 7-DOF beam element introduces warping as an additional degree of freedom. This inclusion furnishes the girder with added stiffness to counteract the applied torsional moment. The results from the 7-DOF beam element, both in terms of deformation and stress, aligned closely with those from the solid element model and experimental data. Thus, a key advantage of employing the 7-DOF beam element is its capability to precisely analyze torsional behaviors without the need for intricate solid element models. This efficiency translates to reduced time spent on numerical modeling.

Another significant benefit of the 7-DOF beam element is its lower computational complexity. As shown in Table 1, the experimental bridge analysis using both the 7-DOF and classical beam elements required significantly fewer nodes and elements compared to the solid element model. The computational demands for the solid element model were substantially higher, reaffirming the time-saving benefits of the 7-DOF model in analyzing complex bridge structures.

**Table 1.** Comparison of numerical models.

|  | **Solid Element Model (SEM)** | **Classical Beam Element Model** | **7-DOF Beam Element Model** |
|---|---|---|---|
| Number of Nodes | 1,348,288 | 6640 (0.49% of the nodes of SEM) | 6640 (0.49% of the nodes of SEM) |
| Number of Elements | 690,152 | 7164 (1.04% of the nodes of SEM) | 7164 (1.04% of the nodes of SEM) |
| DOF at Each Node | 3 | 6 | 7 |
| Global Stiffness Matrix Size | 4,044,864 | 39,840 (0.98% of the nodes of SEM) | 46,480 (1.15% of the nodes of SEM) |

Additionally, the 7-DOF beam element model is compatible with existing finite element programs that use classical beam elements, facilitating its integration. This compatibility means that existing node and element data can be used without the need to generate new data, requiring only modifications to the stiffness matrix and force vector.

For bridge design, particularly in limit state design, understanding internal forces along the girder is essential. Although solid and shell element models provide accurate stress and deflection data, they do not directly yield internal force information for each cross-section, often necessitating additional post-processing. The 7-DOF element model addresses this gap, offering a reliable method for both torsional analysis and the computation of internal forces across every cross-section.

## 7. Conclusions

In this research, the warping effect was included as an addition to the classical beam theory in the kinematics of the beam, and equations for computing the strain of the beam cross-sections were derived. With the application of total potential energy theory, a 7-DOF beam element, which had seven degrees of freedom at each node, was derived. A corresponding finite element computer program was also developed in MATLAB. The following conclusion can be drawn after comparing the 7-DOF beam element model with the solid element model, the classical beam element model, and the field monitoring results from the experimental bridge.

(a) The 7-DOF beam element can accurately evaluate the torsional behaviors of bridge girders under construction load. The results were validated using the results from the solid element model and experimental data.

(b) In terms of stress and vertical deflection, the 7-DOF beam element had the same behavior as the other numerical methods. However, according to the strain equations for the 7-DOF beam, the warping effect can lead to re-distribution of the stress on the cross-section, especially when the torsional moment applied to the structure is significant.

(c) The classical beam element that is widely used in commercial software packages failed to compute the transverse rotation of bridge girders when the torsional moment was applied. The transverse rotation based on the results from the classical beam element was often larger than experimental data due to the lack consideration of warping stiffness during the torsional analysis.

(d) With similar accuracy, the 7-DOF element model requires less modeling effort and computing time compared with the solid element model. For the experimental bridge in this study, the size of global stiffness matrix reduced to 1% when switching the solid element model to the 7-DOF beam element model.

(e) Unlike the solid and shell element models, with which it is difficult to determine the internal forces directly, the 7-DOF beam element can compute the internal force of cross-sections along the bridge girder for design purposes.

(f) The 7-DOF beam element model does not require changes in the nodal and element information of the classical beam element model. Therefore, it is possible to convert a classical beam element model into the 7-DOF beam element without much effort.

(g) The numerical analysis using the 7-DOF beam element can be an alternative approach to the solid element and shell element for bridge analysis, especially when detailed information on stress distribution on the cross-section is not required and internal forces need to be generated.

**Author Contributions:** Experimental design and methodology, L.H., M.A. and R.H.; experimental testing and data analysis, L.H. and M.A.; numerical simulation, L.H. and M.A.; programming, L.H.; writing—original draft preparation, L.H.; writing—review and editing, L.H., M.A. and R.H.; project administration, R.H. All authors have read and agreed to the published version of the manuscript.

**Funding:** This research was funded by the Illinois Department of Transportation, grant number ICT-R27-179.

**Data Availability Statement:** The data presented in this study are available on request from the corresponding author.

**Acknowledgments:** This publication is based on the results of ICT-R27-179, Effectiveness of Exterior Beam Rotation Prevention Systems for Bridge Deck Construction—Phase II. ICT-R27-179 was conducted in cooperation with the Illinois Center for Transportation; the Illinois Department of Transportation, Office of Program Development; and the US Department of Transportation, Federal Highway Administration.

**Conflicts of Interest:** Author Md Ashiquzzaman was employed by the company Ameren Corp. The remaining authors declare that the research was conducted in the absence of any commercial or financial relationships that could be construed as a potential conflict of interest.

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
