# Peer review of "Application and Experimental Validation of Seven-Degree-of-Freedom Beam Element for Girder Bridges during Deck Construction"

_infrastructures, doi:10.3390/infrastructures8120175_

Round 1

Reviewer 1 Report

Comments and Suggestions for Authors

The authors reports an interesting study on a FEM analysis on steel bridge girders during deck construction. Within the study a 7-DOF beam element was used. Results of analysis with the 7-DOF beam element were compared with classical beam model and experimental results. The paper is well structured, the content of appropriate scientific level.

Only minor changes are proposed: 1) Page 3, line 100: There is written 'ABAQUES/CAE'. It must be replaced by 'ABAQUS/CAE'. 2) The number of references is quite low for a research paper and should be enhanced.

Author Response

Thanks for the reviewers' careful review and valuable recommendations. Details of the changes are in the attached file.

Reviewer 2 Report

Comments and Suggestions for Authors

This manuscript reports the application and experimental validation of 7-DOF beam element for girder bridges during deck construction, and it offers many fresh looks.

I recommend publishing this manuscript after the authors revise their manuscript by comprehensively comparing their research with published references on the similar topic, particularly, adding many related references that are not cited in the manuscript, just to name a few:

·       BEAM188 Element, A seventh degree of freedom (warping magnitude), https://www.mm.bme.hu/~gyebro/files/ans_help_v182/ans_elem/Hlp_E_BEAM188.html,

·       Li Hui, Md Ashiquzzaman, NUMERICAL ANALYSIS OF BRIDGE GIRDERS DURING DECK CONSTRUCTION USING 7-DOF FRAME ELEMENT, Journal of Civil Engineering and Technology (JCIET) Volume 9, Issue 2, July– December 2023, pp. 37-44,

·       Claudio Bernuzzi, Marco Simoncelli,The transformation matrix in the 7DOFs beam formulation, Thin-Walled Structures,Volume 190,2023,110951,ISSN 0263-8231.

Comments on the Quality of English Language

double check 

Author Response

Thanks for the reviewers' careful review and valuable recommendations. Details of the changes are in attached file.

Reviewer 3 Report

Comments and Suggestions for Authors

Attached

Comments on the Quality of English Language

Attached

Author Response

Thanks for the reviewers' careful review. Details of the changes are in attached file.
